# Current Understanding in the Clinical Characteristics and Molecular Mechanisms in Different Subtypes of Biliary Atresia

**DOI:** 10.3390/ijms23094841

**Published:** 2022-04-27

**Authors:** Lin He, Patrick Ho Yu Chung, Vincent Chi Hang Lui, Clara Sze Man Tang, Paul Kwong Hang Tam

**Affiliations:** 1Cancer Centre, Faculty of Health Sciences, University of Macau, Macau SAR, China; helin-bolt@outlook.com; 2Division of Paediatric Surgery, Department of Surgery, School of Clinical Medicine, Li Ka Shing Faculty of Medicine, The University of Hong Kong, Hong Kong SAR, China; vchlui@hku.hk (V.C.H.L.); claratang@hku.hk (C.S.M.T.); paultam@hku.hk (P.K.H.T.); 3Faculty of Medicine, Macau University of Science and Technology, Macau SAR, China

**Keywords:** biliary atresia, genetics, neonatal obstructive jaundice, Kasai portoenterostomy, molecular mechanism

## Abstract

Biliary atresia is a severe obliterative cholangiopathy in early infancy that is by far the most common cause of surgical jaundice and the most common indicator for liver transplantation in children. With the advanced knowledge gained from different clinical trials and the development of research models, a more precise clinical classification of BA (i.e., isolated BA (IBA), cystic BA (CBA), syndromic BA (SBA), and cytomegalovirus-associated BA (CMVBA)) is proposed. Different BA subtypes have similar yet distinguishable clinical manifestations. The clinical and etiological heterogeneity leads to dramatically different prognoses; hence, treatment needs to be specific. In this study, we reviewed the clinical characteristics of different BA subtypes and revealed the molecular mechanisms of their developmental contributors. We aimed to highlight the differences among these various subtypes of BA which ultimately contribute to the development of a specific management protocol for each subtype.

## 1. Introduction

Biliary atresia (BA) is a congenital disease characterized by obliterative cholangiopathy. Affected infants experience a varying degree of obstruction in both intrahepatic and extrahepatic bile ducts [1]. During the embryonic stage, the intrahepatic bile ducts and extrahepatic bile ducts originate differently, which suggests that BA is a multifactorial disease with a complex pathogenesis. BA is the leading cause of obstructive jaundice in neonates and the most common indicator for liver transplantation in children worldwide [2].

Traditionally, BA is classified into two forms according to the time of onset: the embryonic form and the perinatal form. Newborns who suffer from the embryonic form develop cholestasis soon after birth and may suffer from concomitant disorders. By contrast, the perinatal form is characterized by a slightly later disease onset, and the majority of patients do not have associated abnormalities. However, with increasing knowledge gained from various clinical research studies, Davenport proposed a new clinical classification algorithm in 2012 [3]. According to his classification, BA is categorized into the following four subtypes: isolated BA (IBA), syndromic BA (SBA), cystic BA (CBA), and cytomegalovirus-associated BA (CMVBA). This novel classification instigates the popularity of the holistic description of the individual features and the investigation of the etiology for the BA subtypes. This study reviews the clinical characteristics of the BA subtypes and describes the molecular mechanisms contributing to their etiology and pathogenesis. In our previous systematic review, we proposed a generalized clinical management protocol for BA based on the current evidence [4]. However, we believe that ideally, different BA subtypes should have a precise and unique management strategy based on different molecular pathways.

## 2. Search Strategy and Inclusion Criteria

We searched English-language articles in the PubMed database using the search terms (isolated biliary atresia) OR (syndromic biliary atresia) OR (cystic biliary atresia) OR (cytomegalovirus-associated biliary atresia). The publications were retrieved on 19 December 2021 and traced back to 1 January 1990. Clinical studies and experimental animal studies that evaluated the molecular mechanism and the treatment efficacy together with the safety profile of monotherapy or polytherapy for the four subtypes of BA met the inclusion criteria. A total of 643 potential papers were obtained for the selection of representative articles to understand the clinical characteristics and molecular mechanisms in different BA subtypes. Apparently, the clinical studies of BA subtypes are scant across the past three decades (Figure 1). We also retrieved relevant clinical studies that had been published or currently ‘in press’ in the ClinicalTrials.gov (accessed on 28 March 2022) database, as stated on 19 January 2022.

## 3. Clinical Characteristics

### 3.1. Phenotypes and Clinical Manifestations of All BA Subtypes

In addition to the difference in embryonic origins, the intrahepatic bile ducts and extrahepatic bile ducts of BA patients have dramatically different structures at the tissue level: Intrahepatic bile ducts are hyperplastic, surrounded by lobules with different degrees of giant multinucleated hepatocytes, and embedded in fibrotic and inflammatory portal tracts; by contrast, extrahepatic bile ducts show segmental or integral loss of the epithelial lining, extensive fibrosis, and occasional inflammatory foci [3]. The common clinical manifestations of all BA subtypes include: (1) progressive infantile jaundice; (2) acholic stool; (3) dark-colored urine; (4) hepatomegaly; and (5) malnutrition.

### 3.2. Surgical Treatment for All BA

The most widely accepted surgical treatment for BA is Kasai portoenterostomy (KPE) which aims to restore biliary drainage from the bile duct to the gastrointestinal tract. It is expected that once the cholestasis is relieved, the liver damage is alleviated and patients can achieve long-term native liver survival (NLS). An early KPE is likely to achieve a higher NLS rate. A 30-year follow-up study published in France reported that the 25-year NLS rate with KPE performed in the first month after birth (38%) was significantly greater than that in the second (27%) and the third months (22%) after birth (*p* = 0.0001) [4].

Adjuvant therapy (AT) for BA includes steroids and choleretic agents. However, there is still no consensus for the treatment duration and the dosage of these drugs. The principle of administering UDCA is based on the fact that UDCA is a hydrophilic bile acid and can reduce other toxic endogenous bile acids to protect the native hepatocytes and cholangiocytes [5].

A single-center prospective phase 3b randomized controlled trial (RCT) from King’s College Hospital (ClinicalTrials.gov (accessed on 28 March 2022) Identifier: NCT00539565) introduced a high dose of prednisolone (5 mg/kg/day) to treat IBA infants who underwent KPE at <70 days after birth and assessed the efficacy of this regimen [6]. The high-dose prednisolone significantly improved 1-month biochemical derangement and 6-month jaundice clearance rate compared with the low-dose prednisolone (2 mg/kg/day), despite no improvement in the 4-year OS and NLS. These promising findings become the basis of the universal practice of administering steroids in the United Kingdom. However, the START trial (ClinicalTrials.gov (accessed on 28 March 2022) Identifier: NCT00294684), a double-blind, placebo-controlled RCT, showed no significant differences in the 6-month bile drainage and the 2-year NLS between BA infants treated with a high-dose steroid and those treated with a placebo. However, the cohort with a high-dose steroid was more likely to have an earlier onset of the first serious adverse event by 1-month post-KPE than the placebo cohort (37.2% vs. 19.0%; *p* = 0.008) [7]. The reason for this contradictory result is in part related to the different participants (i.e., only IBA in the former RCT and all BA subtypes in the latter RCT), suggesting that the clinical subtype could be an important factor associated with the prognostic outcomes of AT with steroids.

### 3.3. Clinical Characteristics of Different BA Subtypes

The most common clinical phenotype of BA is IBA, accounting for 67–89% of all cases (Table 1) [5,8,9,10,11,12,13,14]. A large number of clinical studies have established the close but variable association between CMV infection and BA. For example, several studies tested CMV infection by fluorescence quantitative polymerase chain reaction (PCR) in post-KPE liver biopsies while others tested it by serum IgM antibody. The inconsistency in testing modalities has led to different results among centers [15,16,17,18]. However, because CMVBA is a relatively new emerging subtype, there is a paucity of epidemiological studies that explicitly evaluate this subtype. Davenport et al. reported that CMVBA made up 5–9.5% of their clinical series [5,19]. The remaining cases consisted of CBA and SBA, with the frequency of 5–22.4% and 4.9–10%, respectively (Table 1) [5,8,20,21,22,23]. Of note, different BA subtypes possessed their unique clinical characteristics (Table 1).

IBA: IBA is characterized by the isolated obliteration in bile ducts and later onset of jaundice and cholestasis. Overexpression of hepatic transforming growth factor-b1, collagen, and α-smooth muscle actin is positively correlated with the METAVIR fibrosis stage but can be downsized after an uneventful KPE [24,25]. Interestingly, postoperative IBA infants experience a significantly higher expression of these three proteins than postoperative SBA infants, mapping to more progressive liver fibrosis in patients with the IBA subtype [24,25].

SBA: SBA is considered to be the embryonic form of BA, which occurs in combination with one or more additional defects, e.g., polysplenia syndrome, cat-eye syndrome, esophageal atresia, intestinal malrotation, and dextrocardia.

CBA: A cyst at the portal hepatis of BA infants is the characteristic of CBA. This is occasionally mistaken as another biliary disease—choledochal cyst (CC), which also exhibits abnormal cystic dilatation of the bile duct. It is of paramount importance to differentiate CBA from CC because the treatment and prognosis of these two conditions are entirely different. Some clinical features may help to differentiate CC from CBA: (1) a non-atretic gallbladder; (2) dilated intrahepatic ducts; (3) sludge deposits inside the cysts [26]; and (4) intramural smooth muscle within the cyst wall [27]. Unique imaging and histological features have been reported in CBA, including: (1) positive triangular cord sign (thickness ≥ 4 mm) [28,29]; (2) stable and small cyst size (diameter ≤ 1.5–3.5 cm) [20,26,28,30,31]; (3) gallbladder mucosal irregularity; (4) thickened hepatic artery; and (5) invisible distal common bile duct [28,29]. Laboratory tests demonstrate a significantly increased serum aspartate aminotransferase-to-platelet ratio index (APRI) (at the best cut-off value of 0.73) and total bilirubin (TBIL) (at the best cut-off value of 98.5 mmol/L) in CBA when compared to CC [30]. Because of the discernable cyst at portal hepatis, CBA can be easily detected at an early age and treated with KPE shortly after birth. Thus, CBA infants have the highest jaundice clearance rate and the lowest cholangitis incidence rate among all BA subtypes, and their 5-year NLS is better than those of IBA infants [32,33].

CMVBA: CMV is a double-stranded DNA virus from the Herpesviridae family that can infect and damage bile duct epithelia. CMV infection occurs in approximately 1% of prenatal newborns and 2% of perinatal newborns, but only a very low proportion of CMV-infected neonates may develop CMVBA. The differentiation of CMVBA from other BA subtypes mainly depends on an elevated expression level of CMV-IgM [34]. Furthermore, the CMVBA subtype has significantly higher serum levels of APRi and TBIL, as well as a significantly larger spleen size than other BA subtypes [19]. The prognosis of CMVBA is worse than IBA, with a lower jaundice clearance rate, increased METAVIR fibrosis stage, and more severe biochemical derangement (e.g., a higher APRi value). The NLS and OS are also inferior due to a later KPE [19]. In order to improve the poor prognosis of CMVBA, a polytherapy that incorporates specific antiviral therapy (e.g., intravenous ganciclovir) into the existing treatment (i.e., KPE combined with AT) is recommended. The duration of ganciclovir is guided by the serial measurements of CMV copies. A European multicenter survey suggested that the median time is 1 month [5]. Nevertheless, RCTs are needed to define the optimal treatment duration and dosage.

## 4. Etiologic and Pathogenic Mechanisms in Different Subtypes of BA

With increasing knowledge around BA pathogenesis, scientists started to believe that different BA subtypes may have a unique etiopathogenesis. For example, ciliary dysfunction caused by genetic mutations of *PKD1L1* (i.e., a ciliary calcium signaling-related gene) has been reported to contribute to BA splenic malformation syndrome [35]. Despite the many efforts spent on studying the underlying mechanism that drives BA, our knowledge of this is still limited. In particular, there is a great deficit of knowledge around the precise pathogenesis of different BA subtypes. Common questions raised by scientists are centered around the reasons for the co-existence with other defects or a cyst at the portal hepatis, as well as the different consequences of CMV-infected infants (i.e., health, non-BA cholangitis, IBA, and CMVBA). We herein attempted to summarize the existing knowledge regarding the molecular mechanisms of three contributors, namely ciliary dysgenesis, epithelial injury, and duct obstruction, that have been hypothesized to be responsible for development of different BA variants.

### 4.1. Mechanisms of Ciliary Dysgenesis

In BA liver, the cholangiocyte cilia were found to be less in number and shorter in length. They also exhibit abnormal orientation and movement. Severe chronic cholestasis, due to ductal obliteration, leads to the accumulation of cytotoxic bile acids that damage the cholangiocyte cilia [36]. Two potential susceptibility genes, *MAN1A2* and *ARF6,* participate in the ciliogenesis and planar polarity effector (CPLANE) network [37]. The interaction of CPLANE proteins with the ciliopathy-related protein JBTS17 at the base of cilia recruits intraflagellar transporters. MAN1A2 interacts with ARF6 and EGFR, signaling the pathway to regulate the intrahepatic biliary network development and bile drainage from the liver (Figure 2). *MAN1A2* mRNA and protein expression are significantly lower in BA liver tissue. MAN1A2 dysregulation could induce functional/developmental defects of cilia and contribute to abnormal bile duct development in BA [37,38].

### 4.2. Mechanisms of Epithelial Injury

The immune system plays a central role in the pathogenesis of BA. Overexpression of intercellular cell adhesion molecules on cholangiocytes can promote these cells to produce immunologic costimulatory factors (e.g., B7–1, B7–2, and CD40) that activate antigen-presenting cells (e.g., Kupffer and dendritic cells (DCs)) in portal tracts, followed by the infiltration of activated natural killer (NK) cells, autoimmune T cells (e.g., CD4+ T cells and CD8+ T cells) and helper T cells (e.g., Th1+ cells, Th2+ cells, and Th17+ cells) to the portal tract area [39,40,41]. These NK cells may be immature; in addition to killing virus-infected cholangiocytes [42], these NK cells may lyse uninfected cholangiocytes in the Nkg2d-dependent cells fashion and aggravate the epithelial damage [43,44]. Meanwhile, these infiltrated immune cells release Th-1 cytokines (e.g., *Ifng, Tnfa, Il12p40, Il1a, Mip1, Stat1, Opn*), Th-2 cytokines (e.g., *Il4, Il5, and Il13*), and chemokines (e.g., *Il15, Cxcl2, Cxcl9, Cxcl10, Mcp1, and Mip2*) (Figure 2) [39,45,46,47,48,49]. Overexpression of these cytokines and chemokines is associated with the intrahepatic and extrahepatic bile duct injury, but their expression levels vary among different subtypes of BA patients. For example, no *Opn* overexpression is detected at the interlobular regions and bile duct remnants of SBA [49].

Activation of NK cells by DCs requires the TNFα–TNFR2 axis, and can be accelerated by the viral infection-induced upregulation of *IL-1R1* and *NLRP3* [45,50]. In addition, upregulation of *IL-1R1* and *NLRP3* expression facilitates Th1 cell-mediated release of Th1 cytokines and chemokines (Figure 2) [45]. The peripheral blood of BA infants, especially CMVBA infants, has a very low level of regulatory T cells (Tregs), which play key roles in maintaining peripheral tolerance, preventing autoimmune diseases. Therefore, low Tregs activity in BA infants could lead to reduced auto-immunosuppression, leading to the excessive release of cytokines and exaggerated bile duct injury (Figure 3). Furthermore, RRV infection of *Sat1-/-* mice (failure to mount Th1 cell-mediated immune responses) still causes epithelial injury by Th2 cell-mediated release of Th2 cytokines (e.g., *Il4* and Il13) [46]. There is less lymphocyte and myeloid cell infiltration associated with moderate epithelial damage in RRV-infected *Stat1* and *Il13* double knockout mice, which is in line with findings that both Th1 and Th2 cell-mediated immune responses are involved in RRV-induced bile duct damage [46]. Given the aforementioned findings, experimental studies on animals demonstrate the loss of cytotoxic CD8+ T cells, cytotoxic proteins (perforin and granzyme), and Th1 cell-.

Th2 cell-mediated immune responses, the depletion of NK cells, and the blockade of Nkg2d each prevent severe bile duct epithelial cell damage and bile duct blockage in RRV-infected mice [43,44,46].

Genomic and epigenetic studies showed that single-nucleotide polymorphisms (SNPs), DNA hypomethylation, and abnormal expression of microRNA are associated with epithelial injury of BA. Genome-wide association studies tested the association between BA and common SNPs in specific populations and implicated a number of BA-susceptible genes that may be involved in multifarious actions of BA occurrence (e.g., immunoregulation, gene regulation, inflammatory response, and cell transmigration) (Table 2) [51,52,53,54,55,56,57,58,59,60,61,62,63,64,65,66,67,68,69,70,71,72,73,74,75,76,77,78,79,80]. Interestingly, the associations of SNPs with BA are inconsistent between different populations or among different BA subtypes. For example, SNPs in *EFEMP1* increase the risk of BA for European–American infants, but not for Chinese populations [55,80], and SNP in *PDGFA* (i.e., rs9690350) increases the risk of non-CBA rather than CBA [71]. However, the associations between these SNPs and BA have not been completely confirmed by functional studies; a possible explanation is that the etiology of BA is heterogeneous or may involve nonclassical genetic mechanisms (e.g., DNA methylations and somatic mutations).

DNA hypomethylation (e.g., *LINE-1*, *ALU,* and *SAT2* repetitive sequences and *IFNγ* promoter) triggers IFN-γ-induced epithelial injury in BA infants, which is associated with overexpression of miR-29b and miR-142-5p, causing the downregulation of DNMT1 and DNMT3a/b (Figure 3) [81]. By contrast, hypermethylation of the *FOXP3* promoter in Tregs impairs the Tregs’ immune suppressor function, leading to an aggravated injury of the biliary epithelium (Figure 3) [82]. miR-155 overexpression activates JAK2/STAT3 signaling the pathway to enhance the pro-inflammatory effect and downregulates the expression of the SOCS1 protein, which can suppress chemokine production (e.g., *Cxcl1, Cxcl9,* and *Cxcl10*) [83,84]. The upregulation of miR-200 and the downregulation of miR-124 have complementary roles in promoting cholangiocyte proliferation via the IL-6/STAT3 signaling pathway, and miR-200 upregulation concomitantly inhibits FOXA2 expression that further reinforces the ^a^ risk of BA relative to the minor allele.

The overexpression of IL-6 in cholangiocytes [85]. *FOXA2* is implicated in programming Th2 cell-mediated innate immunity; depletion of FOXA2 incites the recruitment and activation of myeloid dendritic cells and Th2 cells, which results in an increased production of chemokines and Th2 cytokines (Figure 3) [86].

### 4.3. Mechanisms of Duct Obstruction

The mechanisms of bile duct obstruction and those of epithelial injury partially overlap (Figure 2). RRV infection increases the infiltration of liver by IFN-γ-producing CD4+ and CD8+ T cells, leading to the overexpression of Th1 cytokine *Ifng* [47]. Loss of *Ifng* prevents extrahepatic bile duct obstruction, improving long-term survival in RRV-infected newborn mice [47], indicating the key role of *Ifng* in the process of bile duct obstruction. Interestingly, RRV-primed inflammatory obstruction of bile ducts can be reversed by the depletion of IFN-γ-producing CD8+ T cells, but not by that of IFN-γ-producing CD4+ T cells [87]. Moreover, extrahepatic bile duct obstruction is associated with MAN1A2 downregulation in BA infants. MAN1A2 knockdown leads to an accumulation of high mannose-type N-glycans on EGFR that inhibits the tyrosine kinase activity of EGFR [88], which limits the EGFR-dependent branching morphogenesis of the biliary network and has a pro-inflammatory response via increased adhesion of inflammatory cells on extrahepatic bile ducts [89,90]. As such, CD4+ T cells and CD8+ T cells work independently to mediate neonatal bile duct obstruction, suggesting that these factors are potential therapeutic targets to stop the progression of bile duct obstruction.

## 5. Development of Biomarkers to Improve Clinical Managements

The dysregulated genes identified are closely associated with BA etio-pathogenesis via corresponding signaling pathways, and this association provides possible targets for the development of noninvasive biomarkers for the clinical management of BA subtypes. Early diagnosis and prediction of post-KPE prognosis remain the two major challenges. More work is required to advance BA diagnosis such that patients could be better stratified [91]. At present, the diagnosis for BA can only be made postoperatively after an invasive surgical procedure, and there is still a lack of objective parameters to predict the prognosis. Increasing studies are emerging to validate the feasibility of the application of noninvasive biomarkers to address these challenges.

Serum γ-glutamyl transferase (GGT) [92,93,94,95,96] and matrix metallopeptidase-7 (MMP-7) [97,98,99] showed good diagnostic performance in differentiating BA patients from infants with other causes of neonatal cholestasis (non-BA). Our previous systematic review [100] further reaffirmed the high summary sensitivity, specificity, and area under the curve (AUC) of both biomarkers in diagnosing BA. Moreover, serums IL-13 and IL-18 may be the reliable biomarkers to distinguish BA patients from healthy children [101,102,103,104]. The summary sensitivity and specificity of IL-13 calculated by our study were 77% and 85%, respectively [100]. The elevated preoperative serum MMP-7 [97,99,105,106] and postoperative IL-18 [103,104] additionally demonstrated a positive correlation with the severity of post-KPE liver injury (e.g., jaundice and liver fibrosis).

APRi was first proposed by Wai et al. [107] to predict liver fibrosis and cirrhosis in patients with chronic hepatitis C, with the equation: aspartate aminotransferase(AST)/upper value of normal AST/platelet count (10^9^/L) × 100. In the recent decade, several studies have found that postoperative APRi can also predict post-KPE liver fibrosis and cirrhosis of BA infants [108,109,110]. Our results mirrored the findings, with the high summary sensitivity, specificity, and AUC [100]. Postoperative serum hyaluronic acid (HA) [105,111,112] and the *Wisteria floribunda* agglutinin-positive human Mac-2-binding protein (M2BPGi) [110,113] may be the promising biomarkers to predict post-KPE liver impairment, although no available data for the calculation of the summary diagnostic performance have been reported so far.

According to the above findings, we proposed a clinical management protocol that utilizes a series of noninvasive biomarkers to serve as the alternative to replace surgical exploration for the early diagnosis of BA and prediction of post-KPE prognosis [100]. However, a limitation of the management protocol is that the optimal cut-off value of potential biomarkers remains unclear, and no differentiation exists for BA subtypes. The aforementioned molecule markers involved in the three contributors of BA mechanisms may enlarge the arsenal of the management protocol.

## 6. Conclusions

Despite cholestatic jaundice being the common phenotype in different BA subtypes, each subtype has a unique clinical characteristic that can be distinguished. Regardless of the subtype, KPE and appropriate AT remain the most effective treatments. There is an increasing number of clinical trials which aim to optimize the outcome of BA. CBA is most easily diagnosed due to its morphology and carries the best prognosis compared to the other subtypes. On the other hand, CMVBA has the worst prognosis attributed to the possibility of a late KPE, as well as the long-lasting intracorporeal CMV infection. Until now, the specific etiology and pathogenesis for the other BA subtypes remained unclear. In this study, we identified the shared contributors that are known at present, including ciliary dysgenesis, epithelial injury, immune dysfunction, fibrosis, and ductal obliteration. Different molecular mechanisms are involved in the contributors. Further knowledge in these pathways, together with their correlations with clinical features, may open a new avenue to improve the management for different BA subtypes.

## Figures and Tables

**Figure 1 ijms-23-04841-f001:**
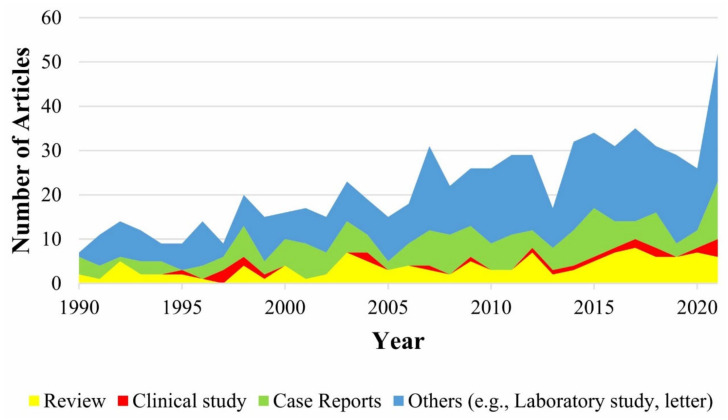
Stacked area chart of different article types regarding the four clinical phenotypes of biliary atresia. Articles were retrieved from the PubMed database ranging from 1 January 1990 to 19 December 2021 using the search terms ((isolated biliary atresia) OR (syndromic biliary atresia) OR (cystic biliary atresia) OR (cytomegalovirus-associated biliary atresia)) AND (mechanism OR treatment OR therapy OR safety).

**Figure 2 ijms-23-04841-f002:**
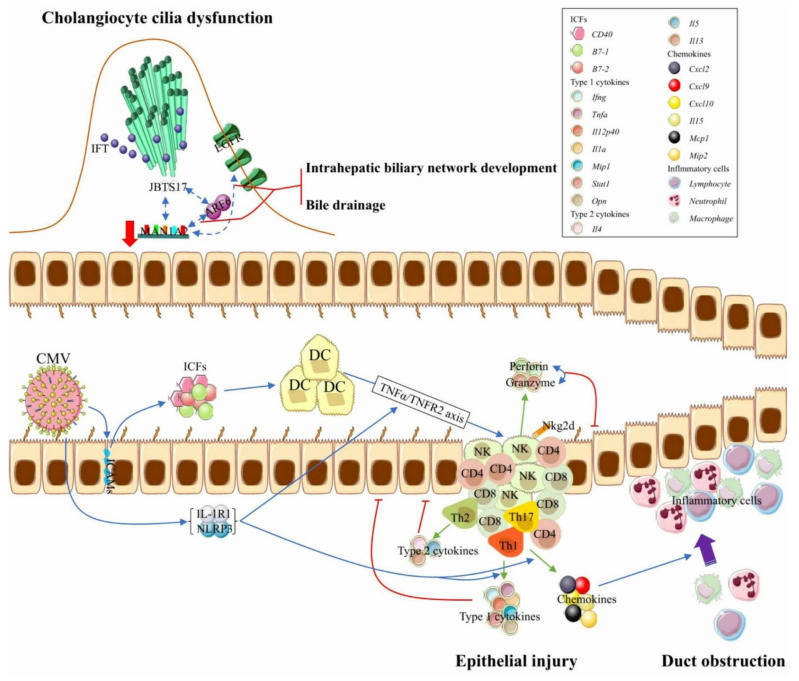
Mechanisms of biliary atresia pathogenesis. The top figure represents the mechanism of cholangiocyte cilia dysfunction, and the bottom figure represents the mechanisms of epithelial injury and duct obstruction. Underexpressed *MAN1A2* and *ARF6* inhibit IFT recruitment and EGFR-mediated biliary network formation. Th1-mediated immune response, Th2-mediated immune response, and the synergistic effect of perforin and granzyme are all responsible for the damage to the bile duct epithelium. Lymphocyte-released chemokines promote the tissue infiltration of inflammatory cells, which leads to intrahepatic and extrahepatic duct obstruction. The inhibition of the TNFα–TNFR axis, the targeted inactivation of *IL-1R1* or *NLRP3*, and the combined loss of *Stat1* and *Il13* each reduce the infiltration of DCs and NK cells, which effectively maintain epithelial integrity and good bile drainage. The solid cyan arrow represents the releasing effect. Abbreviations: IFT, intraflagellar transporter; RRV, rhesus rotavirus; ICFs, immunologic costimulatory factors; DC, dendritic cell; NK, natural killer cell.

**Figure 3 ijms-23-04841-f003:**
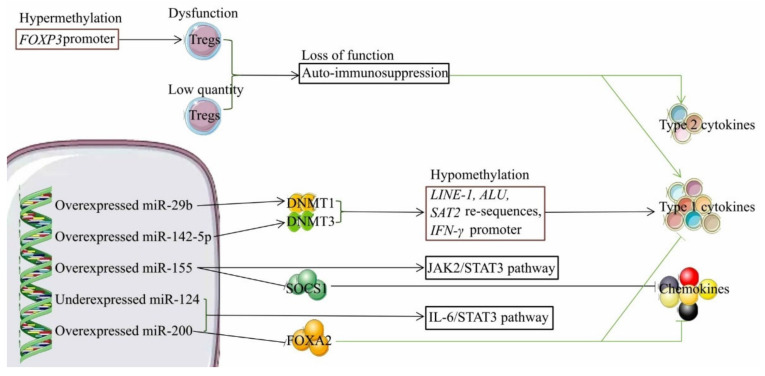
The role of regulatory T cells and dysregulated microRNAs in biliary atresia pathogenesis. Deficits in function and quantity of regulatory T cells contribute to the loss of function in auto-immunosuppression, which gives rise to excessive release of type 1 and 2 cytokines that are both associated with epithelial injury. Downregulated DNMT1 and DNMT3 cause the hypomethylation of *LINE-1*, *ALU*, and *SAT2* repetitive sequences, and *IFN-γ* promoter activity, which triggers type 1 cytokines-induced epithelial injury. The SOCS1 protein can halt chemokine-induced inflammatory cell infiltration that causes bile duct obstruction. The FOXA2 protein arrests the recruitment and activation of dendritic cells and Th2 cells, leading to the low production of chemokines and type 2 cytokines. The solid cyan arrow represents the releasing effect, and the T-shaped solid cyan arrow indicates the inhibition of the releasing effect. Abbreviations: Tregs, regulatory T cells; DNMT1, DNA methyltransferase 1; and DNMT3, DNA methyltransferase 3.

**Table 1 ijms-23-04841-t001:** Clinical classification of BA subtypes.

Clinical Subtype	Distribution Frequency	Characteristic Manifestation	References
IBA	67–89%	Isolated duct obliteration	[7,10,11,12,13,14,15,16]
SBA	4.9–10%	Combination with other defects	[7,10,22,23,24,25]
CBA	5–22.4%	A cyst at the portal hepatis	[7,10,22,23,24,25]
CMVBA	5–9.5%	Cytomegalovirus-IgM positive	[7,21]

Abbreviations: IBA, isolated biliary atresia; SBA, syndromic biliary atresia; CBA, cyst biliary atresia; CMVBA, cytomegalovirus-associated biliary atresia.

**Table 2 ijms-23-04841-t002:** Common single-nucleotide polymorphisms with reported association with biliary atresia.

Gene	3’-UTR Position/Rs Number(Minor Allele)	Population	Risk	Functions	Ref.
*ADD3*	rs17095355 (T)	Chinese	Increased	Cytoskeletal organization	[55,56]
	rs17095355 (T)	Thai	Increased		[57]
	rs17095355 (T)	Thai	Increased		[67]
	rs7099604 (G)	Caucasian	Increased		[58]
*GPC1*	rs6707262 (C) rs6750380 (C)	Chinese	Increased	Growth factors and cytokines	[55]
	rs2292832 (C)	Chinese	Decreased		[59]
*ARF6*	rs3126184 (T)	Chinese	No association	Cytoskeletal organization	[55]
	rs3126184 (T) rs10140366 (C)	Caucasian	Increased		[60]
*EFEMP1*	4 SNPs	Chinese	No association	Cell proliferation	[55]
	rs10865291 (A) rs6761893 (T) rs727878 (T)	Caucasian	Increased		[80]
*USF2*	rs916145 (C)	Chinese	No association	Gene regulation	[61]
	rs916145 (C)	Taiwanese	Increased		[66]
*CD14*	-159 C/T (T)	Taiwanese	Increased	Immunoregulation	[63]
*miR-499*	rs3746444 (G)	Egyptian	Increased	Cell adhesion	[64]
	rs3746444 (G)	Chinese	Increased		[65]
*VEGFA*	rs3025039 (T)	Taiwanese	Decreased	Vasculogenesis, cell proliferation, and cell transmigration	[68]
	rs3025039 (T)	Chinese	No association		[69]
	rs3025039 (T)	Chinese	Decreased		[70]
*PDGFA*	rs9690350 (C)	Chinese	Decreased	Vasculogenesis, cell proliferation, and cell transmigration	[71]
*ADIPOQ*	rs1501299 (T)	Thai	Decreased	Anti-inflammation	[74]
*ITGB2*	rs1160263 (T)	Chinese	Decreased	Immunoregulation	[75]

Abbreviations: ADD3, adducin 3; GPC1, glypican 1; ARF6, ADP-ribosylation factor 6; EFEMP1, EGF containing fibulin-like extracellular matrix protein 1; USF2, upstream transcription factor 2; CD14, cluster of differentiation 14; VEGFA, vascular endothelial growth factor A; *PDGFA*, platelet-derived growth factor subunit A; ADIPOQ, adiponectin; ITGB2, integrin β2.

## Data Availability

Not applicable.

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
