# Peer review of "Current Understanding in the Clinical Characteristics and Molecular Mechanisms in Different Subtypes of Biliary Atresia"

_ijms, 2022, doi:10.3390/ijms23094841_

Round 1

Reviewer 1 Report

The Authors of the paper have reviewed the clinical characteristics of different biliary atresia subtypes and revealed the molecular mechanisms of their developmental contributors. They highlighted the differences among these various subtypes of biliary atresia. This paper is a significant contribution to the scientific discussion about molecular mechanism of biliary atresia pathogenesis and biomarkers to improve clinical managements.

Text editing and minor language revisions should be made. I recommend the paper for publication after minor revision.

Author Response

Thank you for your kind review and encouraging comments.

As suggested, we have performed another round of language to improve the manuscript.  

Please kindly provide your further comment and we look forward to the chance of publication.

Regards

Patrick HY Chung

Reviewer 2 Report

The paper is well described and the sub chapters evaluate well all the clinical aspects. Please consider to report in the " search strategy " the numbers and temporal rate of paper and clinical trial described in your article to give more " power " to the systematic review    

Author Response

Thank you very much for your kind review and the encouraging comments. 

To respond, we have inserted a stacked area chart of different article types regarding the four clinical phenotypes of biliary atresia as a new figure (figure 1). This showed the number and type of articles from different years that were included in this review.  We believe this would enhances the readers' understanding in the articles being evaluated.  The text under 'search strategy' has also been amended in the revised manuscript accordingly.

We hope this will be a satisfactory response and please kindly comment. We look forward to the opportunity to publish.

Regards

Patrick HY Chung
